# Characteristics of Multicomponent Interventions to Treat Childhood Overweight and Obesity in Extremely Cold Climates: A Systematic Review of a Randomized Controlled Trial

**DOI:** 10.3390/ijerph18063098

**Published:** 2021-03-17

**Authors:** Javier Albornoz-Guerrero, Sonia García, Guillermo García Pérez de Sevilla, Igor Cigarroa, Rafael Zapata-Lamana

**Affiliations:** 1Departamento de Educación y Humanidades, Universidad de Magallanes, Punta Arenas 6200000, Chile; Javier.albornoz@umag.cl; 2Faculty of Sport Sciences, Universidad Europea de Madrid, 28670 Madrid, Spain; sonia.garcia@universidadeuropea.es; 3Faculty of Sports Sciences, Department of Physiotherapy, Universidad Europea de Madrid, 28670 Madrid, Spain; guillermo.garcia@universidadeuropea.es; 4Escuela de Kinesiología, Facultad de Salud, Universidad Santo Tomás, Los Ángeles 4440000, Chile; icigarroa@santotomas.cl; 5Escuela de Educación, Universidad de Concepción, Los Ángeles 4440000, Chile

**Keywords:** pediatric obesity, obesity management, treatment, extremely cold climate

## Abstract

Aim: To analyze the characteristics of multicomponent interventions to reduce childhood overweight and obesity in territories with an extremely cold climate. Methods: A systematic review was conducted following the PRISMA statement. MEDLINE, PsycNet, SciELO, and grey literature databases were reviewed in the period between 2010 and 2020. Results: 29 articles were included (*n* = 4434 participants; 9.3 years; 56% women) with an average adherence of 86%, 100% being the highest adherence, for the physical activity and nutrition interventions. The primary variables studied were BMI, BMI Z-score BMI-SDS and, additionally, the secondary variables studied were nutritional status and physical and mental health. In 72% of the interventions presented, positive effects were seen on the reduction of BMI, including in parents and their children. The interventions were carried out mainly by nutritionists in health centers. The duration of the 29 interventions was ≤6 months and ≥12 months, in 59% and 41% of the studies, respectively. 57% of the studies reported post-intervention results. 86% of the interventions included a physical activity component, 80% included a nutrition component, 66% included a behavioral therapy component and 55% included an education component. Concerning the effects of the intervention on the primary outcome, in interventions with a duration equal to or less than six months, the most effective interventions included recreational activities, education, and nutritional programs. In interventions lasting 12 months or more, the most effective interventions included physical activity recommendations, nutritional and physical exercise programs, and cooking classes. Conclusions: This systematic review analyzed the effectiveness of, and characterized, multicomponent interventions lasting for 6 and 12 months, aiming to treat childhood obesity in extremely cold climates. The most frequently used units of measurement were also analyzed and summarized. Evidence derived from RCT. These results can be useful for designing future interventions to treat childhood obesity in territories with an extremely cold climate.

## 1. Introduction

The World Health Organization (WHO) has consistently reported that overweight and obesity are causing a significant deterioration in the health of the world’s population [1]. These conditions have been linked for decades to the development of diseases such as insulin resistance, type 2 diabetes mellitus, high blood pressure, dyslipidemia, and fatty liver disease [2,3,4]. More recently, obesity has been linked to various types of cancer [5], and it has been estimated that 14% and 20% of deaths in men and women, respectively, can be attributed to obesity [6].

Regarding the global prevalence of obesity, it has almost tripled. Specifically, in 2019 the estimation was that 38.2 million children under the age of five were overweight or obese. This is a problem in high-, middle- and low-income countries, particularly in urban settings [7]. Besides, the prevalence of overweight and obesity among children and adolescents aged 5 to 19 years has increased dramatically, similarly among boys and girls [7].

The consequences of childhood obesity are increased school absences and doctor visits [8], so childhood obesity is not simply a risk factor for disease in adulthood, as obese children may experience more illness during childhood [9]. In this regard, early detection and treatment of obesity in children may be the best approach to prevent future increases in morbidity, as well as increases in the health care costs that are likely to occur as overweight and obese children grow older [10,11].

Treatment strategies for childhood obesity have been carried out in multiple ways. Modifying the obesogenic environment and ensuring the acquisition of healthy lifestyle habits has been suggested for more than a decade [12]. Interventions with the family and school are considered as an important strategy against obesity [13]. These strategies consist of nutrition, physical activity, or psychosocial support, and when used in combination they are called multicomponent interventions [14,15].

It has been reported that there is an association between extreme cold weather and childhood overweight or obesity. Thus, it has been shown that cold environments favor the development of obesity since it could influence hormones related to hunger, increasing the appetite towards excessive intake, and promote inactivity [16,17]. Furthermore, this type of climate causes multiple systemic adaptations, such as sympathetic thermogenesis in brown adipose tissue, post-shiver thermogenesis in skeletal muscle, increased production of expired carbon dioxide due to increased metabolic activity, and tachycardia [18]. Similarly, exposure to cold temperatures alters hormonal production, observing an increase in ghrelin and cortisol, which have been associated with an increase in appetite, and facilitating mechanisms of lipid storage [19].

Among the coldest countries in the world are Russia, Canada, USA (north), Iceland, Finland, Estonia, Norway, Sweden, Denmark, Latvia, France, and New Zealand (south) [20]. Scotland in the United Kingdom, the extreme south of Argentina and Chile [21], some areas of the island of Tasmania in Australia, part of the Atlantic coast of Norway, and part of the European islands in the North Atlantic Ocean belong to the subpolar oceanic climate [22]. In this climate, months with an average temperature above 10 °C. are less than four per year.

In Punta Arenas, Chile, winds of 80 miles per hour are common during the southern hemisphere summer. The temperature varies around an average of 6.5 °C, with the lowest temperature of −16.4 °C in Winter, June and July, and the highest of 29.9 °C in summer, January and February [23].

The collected data on extreme cold weather are relevant for designing intervention strategies aimed at reducing overweight or obesity in children in these climates. To date, no systematic reviews have been reported that specify the characteristics of the interventions in children with overweight and obesity in territories with extremely cold climate characteristics.

Therefore, the objective of this review is to analyze the characteristics of multicomponent interventions aimed at reducing overweight and obesity in children in territories with an extremely cold climate.

## 2. Methods

This systematic review was carried out in accordance with the standards established by the PRISMA declaration [24].

### 2.1. Search Strategy to Identify Studies

We reviewed the MEDLINE databases using PubMed, PsycNet, SciELO, and grey literature. The objective was to identify studies that carried out interventions for the treatment of overweight and obesity in children in extremely cold climates. The search covered the period from January 2010 to January 2020 taking as reference the Global Action Plan for the Prevention and Control of NCDs in the Americas. For this search, we used The MeSH terms: “Therapy”, “Therapeutics”, “Rehabilitation”, “Obesity”, “Overweight” and “Child”. The search strategy followed the Peer Review of Electronic Search Strategies (PRESS) [25]. The search terms were: (Therapy OR Therapeutics OR Rehabilitation) AND (Obesity OR Overweight) AND (Child), adapted to each database applying the following filters: period (January 2010 to October 2020), type of article (Randomized Controlled Trial), age (6–12 years) and country (Russia, Canada, United States, Iceland, Finland, Estonia, Norway, Sweden, Denmark, Latvia, France, New Zealand, Scotland, United Kingdom, Argentina, Chile, Faroe Islands, and Iceland).

### 2.2. Study Selection and Inclusion Criteria

Studies that met the following inclusion criteria were selected: (a) Country: interventions carried out in countries with extremely cold climates (Russia, Canada, USA, Iceland, Finland, Estonia, Norway, Sweden, Denmark, Latvia, France, New Zealand, Scotland, United Kingdom, Argentina, Chile, Faroe Islands, and Iceland); (b) Sample: boys and girls between 6 and 12 years old; (c) Methodological Design: randomized controlled clinical trial; (d) Period: between 2010 and 2020. Exclusion criteria were: reviews, editorial documents, protocols, and doctoral thesis (Table 1).

### 2.3. Data Extraction

First, we used the Mendeley tool to look for duplicates. As no duplicates were found, then the articles that met the inclusion criteria were selected. When decisions could not be made considering only the title and abstract of the article, the full text was retrieved (Figure 1). We used a standardized questionnaire to extract the data from the included articles to synthesize the evidence. The information extracted included: (a) general characteristics of the studies and the participants (author, year, country, sample size, adherence, reasons for withdrawal, recruitment, age, sex, inclusion criteria, and place of intervention) (b) main characteristics of the interventions; (c) characteristics of the interventions according to the component addressed; (d) main variables assessed (physical health and mental health); (e) main assessment instruments used.

### 2.4. Risk of Bias Tool

To assess the methodological quality of the studies included in this review, we used the “Cochrane Handbook of Systematic Reviews of Interventions” [26], evaluating the risk of bias in each of the proposed items: (a) Selection bias, (b) Performance bias, (c) Detection bias, (d) Attrition bias and (e) Reporting bias. The only item that was not considered was “Other biases”. The results of this analysis are presented, grouping all the studies analyzed (Figure 2).

### 2.5. Data Synthesis Strategy

The main information of the included studies is presented in summary tables and figures. In the discussion, the most relevant methodological and applicability aspects are analyzed and some suggestions for future research are given.

## 3. Results

### 3.1. Search Outcome

Figure 1 shows the flow chart made following the PRISMA Statement. We identified 1056 potential studies on the treatment of childhood overweight and obesity in countries with extremely cold climates. After the exclusion of the duplicates in the databases, the screening and eligibility criteria were applied. Finally, 29 articles were included for data synthesis in this review [27,28,29,30,31,32,33,34,35,36,37,38,39,40,41,42,43,44,45,46,47,48,49,50,51,52,53,54,55].

### 3.2. Characteristics of the Studies Analyzed

29 articles were included in this review, whose interventions were carried out in the following countries considered as extreme cold climate zones: Sweden, Canada, Denmark, United States, Iceland, Germany, Norway, Switzerland, England, Finland, Holland, and China. Considering all the studies, the total number of subjects at baseline was 4434, with an average adherence of 86%. The adherence to the programs, depending on the type of the intervention, was: (1) Physical activity and nutrition: 100%; (2) education: 93%; (3) behavioral and education: 93%; (4) physical activity and education and nutrition: 87%; (5) behavioral and nutrition: 86%; (6) physical activity and behavioral: 86%; (7) physical activity and behavioral and education and nutrition: 84%; (8) physical activity and behavioral and nutrition: 81%. Of these studies, 93% reported losses in their sample, the main reasons being categorized as follows: lack of time (10%), personal reasons (24%), attendance (41%), health (20%), not meeting criteria during the development of the intervention (17%), did not wish to continue (27%) and unspecified (34%). 58% of the studies recruited children from health centers, 27% from educational centers, and 51% from the community. The mean age was 9.3 years. Concerning gender distribution, 56% were women and 44% men. Regarding the prevalence of diseases, only two studies detailed this information: one included a child with asthma, and the other only healthy children. The main inclusion criterion (90%) was the BMI value corresponding to overweight or obesity, while in 3% of the studies it was the value of weight for height and 7% did not specify this information. 51% of the interventions were carried out in health centers, 13% in educational centers, 13% in university centers, and 13% in community centers. Of these studies, 13% did not specify this information (Table 2).

### 3.3. Risk of Bias Assessment

The risk of bias analysis revealed that the distribution of biases classified as “low risk” or “unclear risk” was similar, except for detection bias, in which 66% of the studies presented “unclear risk”, and reporting bias, in which 100% of the studies presented “low risk”. Among the four types of bias, studies classified as “high risk” were only for “performance bias”, where more than 10% (22%) of the studies presented “high risk” (Figure 2).

### 3.4. Main Characteristics of the Interventions

The studies presented as main result six variables related to childhood overweight and obesity: 38% reported BMI z-score, 34% reported BMI-SDS (standard deviation), 24% considered BMI (kg/m2) and 10% considered other variables, these being the percentage of body fat, body weight according to height and BMI-p (percentile). Regarding the main post-intervention result, 72% of the interventions presented significant changes, while 28% did not present significant post-intervention changes. Only nine studies analyzed the maintenance of this effect over time, obtaining statistically significant maintenance in 77% of cases. One of the inclusion criteria of 38% of the studies was that at least one of the parents was overweight or obese. In relation to the subjects who underwent the intervention, 4% included only parents, 6% only children, 52% included parents and children, and 34% included the entire family. In 59% of the studies, the duration of the intervention was equal to or less than 6 months, and in 41% it was equal to or greater than 12 months. 57% of the studies presented follow-up of post-intervention results. Of these, 43% followed up for a period equal to or less than 6 months, while 57% followed up for a period equal to or greater than 12 months. Regarding the professionals who participated in the interventions, 52% were nutritionists, 34% qualified professionals in exercise prescription (physiotherapists, physical education teachers, and personal trainers), 32% doctors, 27% psychologists, 27% other types of professional (interventionists, counselors or facilitators), 23% included a nurse, 2% a social worker and 9% did not specify this information. The interventions were based on four components: (1) physical activity, (2) education, (3) behavioral therapy, (4) and nutrition. The physical activity component was included in 86% of the interventions, being implemented in 55% of cases as recommendations (general exercise recommendations, promotion of an active life, and reduction of time spent in sedentary behaviors), in 16% of cases as physical exercise programs (structured training program including mainly aerobic exercise, circuit training, and spinning), and in 23% of cases as recreational activities (mainly games). Of the interventions. 55% included an education component, which was implemented mainly as healthy lifestyle recommendations focused on health promotion. Regarding behavioral therapy, 66% of the interventions included this component through cognitive-behavioral strategies: stimulus control, self-monitoring, coping strategies, dealing with problems, positive reinforcement, planning, goal setting, behavior skills, and parenting skills. Finally, 80% of the interventions included a nutrition component, which was implemented in 45% of cases as recommendations focused on healthy eating (such as increasing the intake of water, fruits, and vegetables and reducing the intake of high-calorie foods), in 34% of the cases as food plans using mainly the “traffic light system”, and in 6% of cases healthy eating workshops were included (Table 3).

### 3.5. Effects of the Interventions on the Primary Outcome According to the Duration of the Intervention

Duration equal to or less than six months: Considering all types of interventions, 77% obtained significant effects for the primary outcome. All the interventions that included the physical activity component showed significant effects on the primary outcome with the recreational activities modality, while with the recommendations modality they obtained significant effects in 59% of cases, and 83% of cases using the physical exercise program modality. All the interventions that included education had significant effects on the primary outcome, whereas those that included behavioral therapy had significant effects in 65% of cases. Finally, 100% of the interventions that included a nutrition component showed significant effects on the primary outcome when using the diet program modality, 63% with the recommendation’s modality, and 50% when using the healthy cooking workshop modality. Furthermore, considering that all the interventions included at least two components, they all presented significant effects when they included the physical activity component in the form of recreational activities. However, the interventions that combined physical activity (recommendations approach), behavioral therapy, and nutrition component (recommendations approach) only presented significant effects when they also included the education component,

Duration equal or more than twelve months: of the interventions, 89% had significant effects on the primary outcome. 67% of interventions that included a physical activity component presented significant effects with the recreational activities modality, 100% with the recommendations modality, and 100% with the physical exercise program modality. Regarding interventions that included education, 92% presented significant effects on the result. The interventions that included behavioral therapy had significant effects in 91% of cases. Finally, 100% of the interventions that included a nutrition component showed significant effects on the primary outcome with the diet program modality, 75% with the recommendations modality, and 100% with the healthy cooking workshop modality. Furthermore, only two interventions (11%) did not present significant effects, having in common that both included the nutrition component in the modality of recommendations (Table 4).

### 3.6. Variables Studied in the Articles

The variables studied in the different studies were grouped into three categories: nutritional status, physical and health condition, and psychological variables.

#### 3.6.1. Nutritional Status

This category was considered in 100% of the studies and included eight variables: BMI z-score (86% of studies), BMI (79%), waist circumference (44%), body composition (mainly fat mass and lean mass; 34%), waist-to-height ratio (14%), skinfold (10%), hip circumference (6%) and neck circumference (3%) (Figure 3a).

#### 3.6.2. Physical and Health Condition

This category was considered in 76% of the studies and included eight variables: physical activity level (48% of studies), food intake (41%), blood pressure (34%), health biomarkers (31%), sedentary behavior (20%), aerobic fitness (17%), eating behavior (10%), muscular strength (3%) (Figure 3b).

#### 3.6.3. Mental Health

This category was considered in 24% of the studies and included seven variables: health-related quality of life (17% of studies), self-perception (6%), self-esteem (6%), mood (3%), anxiety (3%), depression (3%), and strengths and difficulties perceived by parents (3%) (Figure 3c).

### 3.7. Assessment Instruments

The instruments used in the studies were grouped into three categories: nutritional status, physical and health condition, and psychological variables.

#### 3.7.1. Nutritional Status

To assess these variables, six instruments were used: tape measure (34% of studies), anthropometry measurement (17%), bio impedance measurement (13%), dual-energy x-ray absorptiometry (DXA) (13%), caliper (6%), and the three-component model of body composition (3%) (Figure 4a).

#### 3.7.2. Physical and Health Condition

To assess these variables, 20 instruments were used, and were sub-grouped into four categories:

Nutrition: to assess this category, the instruments used were food frequency questionnaire (17% of the studies), 3-day diet recall form (10%), 24h-dietary recall (3%), Block Kids questionnaire (3%), Dutch eating behavior questionnaire for children (DEBQ-C) (3%), The Family Eating and Activity Habits questionnaire (FEAH) (3%), Day in the life questionnaire (DiLQ) (3%) and 2-day nutritional dietary record (3%) (Figure 4b).

Level of physical activity and sedentary behavior: to assess this category, the instruments used were non-validated questionnaire for dietary, physical activity and sedentary behaviors (33% of studies), accelerometers (13%), pedometer (6%), Physical activity questionnaire for children (PAQ-Q) (6%), Leisure Score Index (LSI) (3%) (Figure 4c).

Physical condition: to assess this category, the instruments used were walking test (6% of studies), handgrip strength test (3%), ergo-spirometry test (3%), jumping test (3%) (Figure 4d).

Cardiovascular risk: 31% of the studies took blood samples, while 13% measured blood pressure with a manual blood pressure cuff, and 20% with an automatic sphygmomanometer (Figure 4e).

#### 3.7.3. Psychological Variables

Ten instruments were used to assess these variables, and were sub-grouped into two categories:

Psychological health: to assess this category, 6% of studies used the self-perceptions profile questionnaire for children (SPPC), 6% the children’s depression scale (CDS), 6% the questionnaire of strengths and difficulties of the child reported by the parents (SDQ), 3% used the child and youth physical self-perception profile questionnaire (CY-PSPP), 3% the scale of intrinsic versus extrinsic orientation in the classroom, and 3% the multidimensional anxiety scale for children (MASC) (Figure 4f).

Health-related quality of life: of the studies, 6% used the pediatric quality of life questionnaire (PedsQL), 3% the child health questionnaire, Dutch version (CHQ-PF50) and 3% the quality of life questionnaire for children between 7 and 13 years old (KID-KINDL-R) (Figure 4g).

## 4. Discussion

The main finding of this review is that interventions that are effective for the management of childhood overweight and obesity are those that combine the components of physical activity, diet, education, and behavioral therapy, and that involve children and parents as participants in the intervention. Specifically, the most effective interventions with a duration equal to or less than six months included recreational activities, education, and nutritional programs. The most effective interventions lasting 12 months or more included physical activity recommendations, nutrition programs, physical exercise programs, and cooking classes.

For more details, the findings of this review are organized into four categories: the general characteristics of the studies, the assessment of the risk of bias, the main characteristics of the interventions, and the variables studied, as well as the assessment instruments used. Their importance and possible implications are discussed below.

### 4.1. General Characteristics of the Studies

In this review, 12 countries belonging to North America, Europe, and Asia were included, where interventions were developed for the management of obesity in children between 6 and 12 years of age with overweight, or obesity in geographic areas with extremely cold climates. Childhood overweight and obesity are a global problem [1,7], with a higher prevalence in regions where temperatures are lower [16,18]. Although 93% of the studies suffered losses in their sample, the average adherence was very high, 86%, which makes for highly relevant data, since the results of an intervention are dependent on the adherence of its participants. This is in contrast to what some authors state, that dropout at 6 months is close to 40% [56,57]. This high adherence could be related to the fact that the interventions had a large participation of parents, and that most of them included a behavioral therapy intervention, which is proposed as an effective tool to improve adherence to management programs of childhood obesity [58]. Regarding recruitment, this was carried out mainly from health centers. This could be related to the availability of the records in the health systems of developed countries, which in turn allow better access to potential participants. The main inclusion criterion was BMI (90%), surpassing other measures that could be more representative of overweight and obesity, such as the percentage of body fat. BMI, even though it is less specific, is the main tool for classifying overweight and obesity, perhaps because it is easy to access and low cost. BMI is the indicator recommended by the WHO to assess anthropometrically the population under 20 years of age, although it is essential to consider the degree of pubertal development before applying it [59,60]. 50% of the interventions took place in health centers, and only 13% in educational centers, although educational establishments are the space in which children spend a large part of their time during the day

### 4.2. Risk of Bias Assessment

The distribution of biases classified as “low risk” or “unclear risk” was similar, except for detection bias, which presented a 66% “unclear” risk. Detection bias is related to the blinding of the assessors to the study results, and although the Cochrane manual states that their blinding does not ensure success, lack of blinding could bias the study results [26]. In any case, the predominant classification of this type of bias, in particular, was “unclear risk” and not “high risk”, which is associated with a lack of information regarding this bias on the part of the authors, rather than a possible bias in the results. Besides, although the studies had some “high risk” bias classification, the highest proportion was present in performance bias with 22%, and less than 10% in the other types of bias. This could be because all the studies have an experimental and randomized methodological design (RCT), which corresponds to the highest level of evidence in terms of quantitative studies.

### 4.3. Main Characteristics of the Interventions

Most of the studies considered BMI as the main outcome to assess the effects of the interventions. Although this is logical, since it was the main inclusion criterion to classify overweight and obesity, some authors question its use because it is less sensitive to weight change in children with higher body weight [61]. A change in BMI does not necessarily imply a decrease in body fat [62], but it is the indicator recommended by the WHO for anthropometric evaluation in children [59,60]. Most of the studies considered the participation of at least one parent during the intervention. Knowing the role that parents play in the development of obesity in their children, it is of great value that they can actively participate in interventions for its management [63,64]. The current gold standard treatment for the management of overweight and obesity in childhood is aimed at both children and parents [65,66], combining components such as physical activity, diet, education, and behavioral therapy [67]. Combined treatment, in addition to nutritional and physical activity intervention, should include behavioral management and education [68,69], to guarantee the maintenance of long-term effects through changes in behavior that promote sustained changes in lifestyle [58,70]. The effects of interventions that combine components have been documented both in the short term (≤6 months) and in the long term (≥12 months) [69,71]. The interventions that lasted 6 months or less presented significant effects in 77% of cases, and according to the components addressed the modalities that presented greater effectiveness were: in physical activity, recreational activities (100%), nutrition and diet program (100%), education (100%) and behavioral therapy (65%). Regarding the interventions that had a duration of at least 12 months, 89% presented significant changes in their primary effect, and according to the components addressed the modalities that presented greater effectiveness were: physical activity recommendations (100%), the modality of diet programs (100%), and the components of education (92%) and behavioral therapy (91%). It seems that the modality of physical activity recommendations, which consists of recommendations of intensity, time, and frequency, in addition to promoting active transport and reducing sedentary behaviors, has an important effect when the interventions are long-term, but not when the duration of the intervention is less than or equal to 6 months. Regarding the nutrition component, the healthy eating recommendations mode does not have as powerful an effect as the nutrition plans mode. Furthermore, the interventions that include the behavioral therapy component appear not to be as effective when combined with physical activity recommendations and healthy eating recommendations, as opposed to when combined with recreational activities and nutrition plans. It is important to emphasize that the treatment of obesity is not focused only on the reduction of body weight, and it is advisable to do a post-intervention follow-up to maintain the long-term effects [72]. Despite this, only 31% of the studies considered a post-intervention follow-up evaluation

### 4.4. Variables Studied and Assessment Instruments Used

The nutritional status variables were present in all studies, with BMI being the most studied variable despite its limitations, such as not allowing the distinction between fat mass and lean mass. It has also been observed that some children with normal BMI have fat mass values in the obesity range [73]. Only about 40% of the studies evaluated body composition through variables other than BMI. To do this, they used doubly indirect methods such as waist circumference and indirect methods such as bioimpedance and DXA, which some authors propose as a better alternative [74]. The physical health variables were considered in 76% of studies, with the level of physical activity being the most analyzed variable, recorded mainly by indirect methods such as questionnaires, with direct methods such as pedometers and accelerometers rarely used. This could be related to the fact that the physical activity component was approached mainly in the form of physical activity recommendations, rather than with physical exercise programs. Psychological variables were assessed in only 24% of the studies, with health-related quality of life being the most analyzed variable through questionnaires, surpassing other variables such as depression, anxiety, and self-esteem. Obesity is a disorder of multifactorial origin that affects multiple areas of human development, and we believe that the evaluation of psychological variables can provide very relevant data, since children with obesity have a very deteriorated image of themselves, feeling inferior and rejected, which is a drag on their social and psychological development in the short and long term [75,76].

## 5. Limitations and Contributions

A limitation of this study is not having found, in the literature, interventions carried out in other representative countries in areas of extreme cold climate such as Russia, or southern Chile and Argentina. However, studies from 12 countries on three different continents were included, allowing a broad view of the characteristics of interventions to combat childhood overweight and obesity in geographic areas of extreme cold climate. This study can serve as a guide to implement interventions in areas with these characteristics, knowing which ones are the most effective, concerning the components addressed and their duration.

## 6. Conclusions

The interventions that were effective in the management of childhood overweight and obesity were those that combined the components of physical activity, diet, education, and behavioral therapy and that involved both children and their parents as participants in the interventions. In interventions with a duration equal to or less than six months, the most effective included recreational activities, education and nutritional programs. In interventions lasting 12 months or more, the most effective included recommendations for physical activity, nutritional and physical exercise programs, and cooking classes. This demonstrates that the effectiveness of these programs requires a combination of components, with a multi-professional approach, aimed at implementing strategies that favor a healthy lifestyle in children, involving parents as active participants in the process. The results of this review serve as a reference to design future effective interventions in the management of childhood obesity in extremely cold climates.

## Figures and Tables

**Figure 1 ijerph-18-03098-f001:**
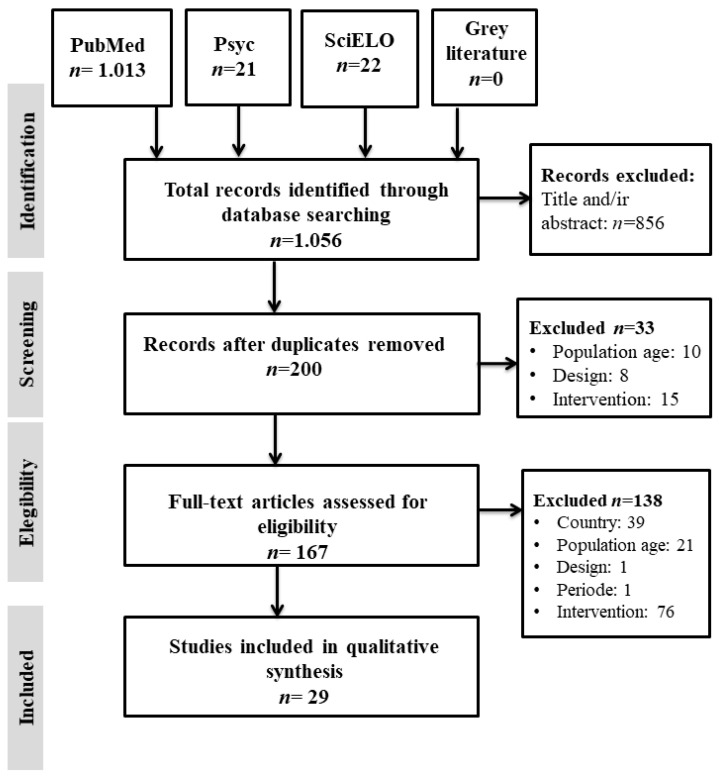
PRISMA flow chart.

**Figure 2 ijerph-18-03098-f002:**
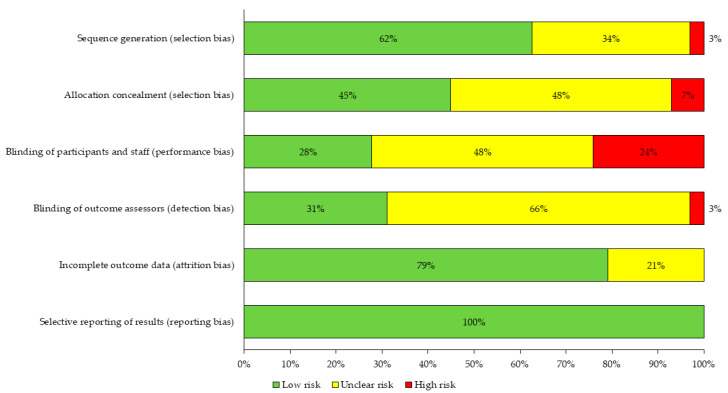
Risk of bias assessment.

**Figure 3 ijerph-18-03098-f003:**
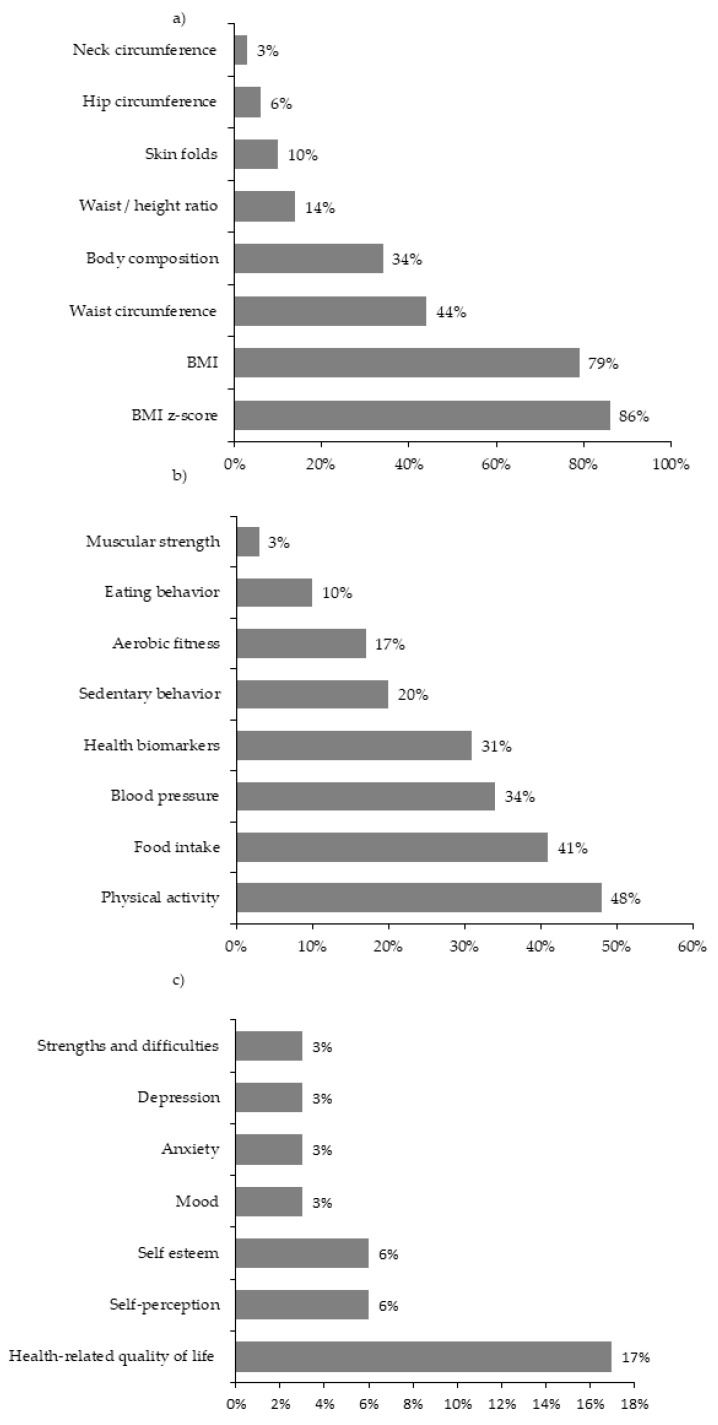
Variables of (**a**) nutritional status, (**b**) physical and health condition, and (**c**) psychological health.

**Figure 4 ijerph-18-03098-f004:**
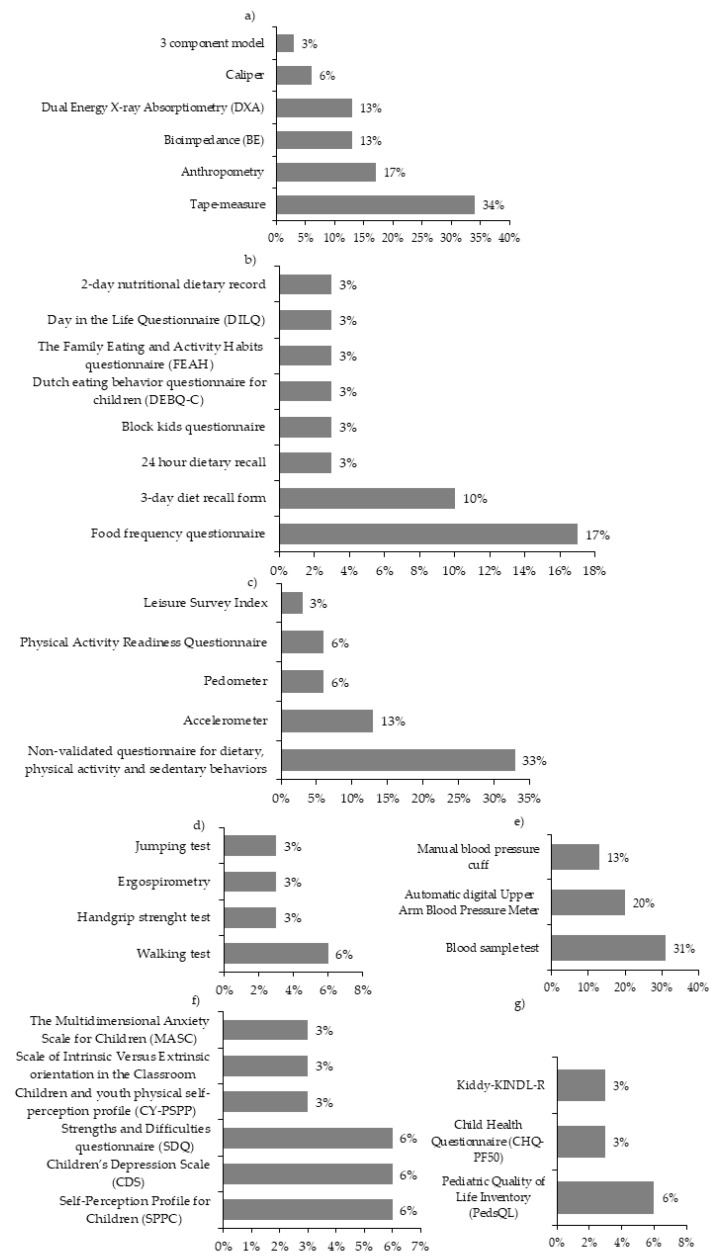
Instruments for the assessment of (**a**) nutritional status, physical and health condition; (**b**) nutrition, (**c**) level of physical activity and sedentary behavior, (**d**) physical condition, (**e**) cardiovascular risk, psychological variables; (**f**) health and (**g**) health-related quality of life.

**Table 1 ijerph-18-03098-t001:** Eligibility criteria of the studies.

Criteria	Description
(1) Type of intervention: multi-component intervention Duration	(a) Education, behavioral therapy, nutrition therapy, and physical activity or physical exercise (b) To present complementary interventions (more than one) (c) At least four weeks
(2) Participants	(a) Overweight or obese; boys and girls; aged 6 to 12 years
(3) Study variables for intervention effects	(a) Physical health: aerobic fitness, muscular strength, flexibility, biomechanics of gait, balance, proprioception or others. (b) Mental health: depression, mood, self-esteem, health-related quality of life, anxiety, or others. (c) Cognitive skills: memory, perception, language, attention, concentration, or others
(4) Type of article	(a) Original article with experimental design and random assignment.
(5) Country of origin of the population	(a) Countries with extremely cold climate

**Table 2 ijerph-18-03098-t002:** Characteristics of the studies analyzed in multicomponent interventions in extremely cold climates.

Ref.	Main Author, Year, Contry	Sample Size	Adherence	Reason for Exclusion							Recruiting			Age	Sex		Inclusion Criteria			Place of the Intervention				
															♀	♂								
			%	LT	PR	AT	NS	HE	DNMC	DWC	HC	EC	COM	mean	%	%	BMI	W/H	NS	HC	EC	UC	CC	NS
[27]	Mårild S. (2013), Sweden	265	93	-	-	-	1	-	-	-	-	1	-	11	58	42	-	-	1	1	-	-	-	-
[28]	Cohen T. (2016) Canada	78	93	1	1	-	-	-	-	-	1	1	1	8	58	42	1	-	-	-	-	1	-	-
[29]	Traberg C. (2016) Denmark	115	75	-	-	-	1	1	1	1	-	1	-	12	45	55	1	-	-	-	-	-	1	-
[30]	Saelens B. (2013) USA	89	80	-	-	1	-	-	-	-	1	-	1	9	67	33	1	-	-	-	-	-	-	1
[31]	Gunnarsdottir T. (2014) Iceland	16	81	-	1	1	-	-	-	1	1	-	-	10	68	32	1	-	-	1	-	-	-	-
[32]	Harder-Lauridsen N. (2014) Denmark	38	92	-	1	-	-	-	1	-	-	1	-	8	80	20	1	-	-	-	1	-	1	-
[33]	Reinehr T. (2010) Germany	66	91	-	-	-	1	-	-	-	1	-	1	11	58	42	1	-	-	-	1	-	-	-
[34]	Benestad B. (2017) Norway	94	73	-	1	1	-	-	1	1	1	-	-	10	50	50	1	-	-	1	-	-	-	-
[35]	Danielsson P. (2016) Sweden	589	100	-	-	-	-	-	-	-	1	-	-	9	46	54	1	-	-	1	-	-	-	-
[36]	Epstein L. (2014) USA	101	66	-	-	-	1	-	-	-	-	-	1	11	60	40	1	-	-	-	-	1	-	-
[37]	Wilfley D. (2017) USA	172	93	-	-	-	1	-	-	-	-	-	1	9	61	39	1	-	-	-	-	1	-	-
[38]	Larsen L. (2015) Denmark	80	93	-	-	-	1	-	-	-	-	-	1	6	64	36	-	-	1	-	-	-	-	1
[39]	Farpour-Lambert N. (2019) Switzerland	74	89	-	-	-	-	1	-	1	1	-	-	9	51	49	1	-	-	1	-	-	-	-
[40]	Wylie-Rosett J. (2018) USA	366	73	-	-	1	-	-	1	1	1	-	-	9	52	48	1	-	-	1	-	-	-	-
[41]	Warschburger P. (2016) Germany	685	76	-	-	-	1	-	1	-	1	-	-	11	52	48	1	-	-	1	-	-	-	-
[42]	Christison A. (2016) USA	84	95	-	-	1	-	-	-	-	1	-	1	10	54	46	1	-	-	-	-	-	1	-
[43]	Croker H. (2012) England	72	79	-	-	1	-	1	-	1	1	1	1	10	70	30	1	-	-	1	-	-	-	-
[44]	Kokkvoll A. (2015) Norway	97	85	-	1	1	-	1	-	-	-	-	1	10	54	46	1	-	-	1	-	-	-	-
[45]	Doyle-Baker P. (2011) Canada	27	93	-	-	1	-	-	-	-	-	-	1	8	52	48	1	-	-	-	-	-	-	1
[46]	Kalavainen M. (2012) Finland	70	97	-	-	-	-	-	-	-	-	-	1	8	60	40	-	1	-	1	-	-	-	-
[47]	Njardvik U. (2018) Iceland	90	67	-	-	-	1	-	-	-	-	1	-	11	45	55	1	-	-	1	-	-	-	-
[48]	Gerards S. (2015) The Netherlands	86	85	1	1	1	-	-	-	-	1	-	1	7	55	45	1	-	-	1	-	-	-	-
[49]	Boutelle K. (2011) USA	80	63	1	1	-	-	1	-	1	1	-	1	10	60	40	1	-	-	-	-	1	-	-
[50]	Robertson W. (2017) England	105	84	-	-	1	-	1	-	1	1	1	1	9	65	35	1	-	-	-	1	-	1	-
[51]	Ming Hao M. (2019) China	229	100	-	-	-	-	-	-	-	-	1	-	10	45	55	1	-	-	-	1	-	-	-
[52]	Steven D. (2014) USA	72	97	-	-	1	-	-	-	-	1	-	-	6	36	64	1	-	-	1	-	-	-	-
[53]	de Niet J. (2012) Nederlands	144	98	-	-	-	1	-	-	-	1	-	-	10	64	36	1	-	-	1	-	-	-	-
[54]	Sherwood N. (2018) USA	421	86	-	-	1	-	-	-	-	1	-	-	6	50	50	1	-	-	1	-	-	-	-
[55]	Saelens B. (2017) USA	29	82	-	-	-	1	-	-	-	-	-	1	10	59	41	1	-	-	-	-	-	-	1

Ref. reference number, 1: specified data, (-): No specified data, **Reason for exclusion:** LT, Lack of time; PR, Personal reasons; AT, Attendance; HE, Health; DNMC, Does not meet the criteria; DWC, Does not want to continue. **Empty box**: Does not meet the criteria. **Recruiting:** HC, Health center; EC, Educational center; COM, Community. **Inclusion criteria:** MBI, Body mass index; W/H, Weight to height; NS, No specified. **Intervention place:** HC, Health center; EC, Educational center; UC, University center; CC, Community center.

**Table 3 ijerph-18-03098-t003:** Main characteristics of multicomponent interventions according to component addressed.

Ref.	Group	Participants				Interventions Characteristics										Intervention								Primary Outcome			Main Outcome		
						Dur. Months	Tra	Professional Responsible								Phys. Act			Education	Beh. Ter	Nutrition			BMI	BMI	BMI			
		P	C	P-C	FAM			Nur	Nut	PA	MD	Psy	SW	Other	NS	Rec	PEP	RA	Rec	Cond	Rec	NP	RA	(kg/t2)	(z-)	(SD)	SC	MT	NSC
[27]	IG1	-	-	1	-	12	-	1	1	-	-	-	-	-	-	1	-	-	-	1	1	-	-	-	1	-	1	-	-
	IG2	-	-	1	-	12	-	1	1	1	-	-	-	-	-	1	-	-	-	1	1	-	-	-	-	-	-	-	-
[28]	IG1	-	-	-	1	12	-	-	1	-	-	-	-	-	-	1	-	-	1	-	-	1	-	-	1	-	1	-	-
	IG2	-	-	-	1	12	-	-	1	-	-	-	-	-	-	1	-	-	1	-	-	1	-	-	-	-	-	-	-
[29]	IG1	-	-	-	1	1.5	1	1	1	-	-	-	-	-	-	-	-	1	-	1	-	1	-	1	-	-	1	-	-
	IG2	-	-	1	-	1.5	1	1	1	1	-	-	-	-	-	-	1	-	1	-	-	-		-	-	-	-	-	-
[30]	IG1	-	-	1	-	5	1	-	-	-	-	-	-	1	-	1	-	-	-	1	-	1	-	-	1	-	1	1	-
	IG2	-	-	1	-	5	1	-	-	-	-	-	-	1	-	1	-	-	-	1	-	1	-	-	-	-	-	-	-
[31]	IG	-	-	1	-	4	1	-	-	-	-	-	-	-	-	1	-	-	1	1	-	1	-	-	-	1	1	1	-
[32]	IG	-	-	-	1	5	-	-	1	1	-	-	-	-	-	-	1	-	1	-	1	-	-	1	-	-	1	-	-
[33]	IG	-	-	-	1	6	-	-	1	1	1	1	-	-	-	-	-	-	1	-	-	1	-	1	-	-	1	-	-
[34]	IG1	-	-	-	1	6	1	1	-	1	1	-	-	-	-	1	-	-	-	1	-	-	-	1	-	1	-	-	1
	IG2	-	-	-	1	6	1	1	1	1	1	1	-	-	-	1	-	-	-	1	1	-	-	-	-	-	-	-	-
[35]	IG	-	-	1	-	60	-	1	1	1	1	1	-	-	-	-	1	-	1	-	1	-	-	-	-	1	1	1	-
[36]	IG1	-	-	1	-	12	-	-	-	-	-	-	-	-	1	1	-	-	-	1	-	1	-	1	-	-	1	-	-
	IG2	-	-	1	-	12	-	-	-	-	-	-	-	-	1	1	-	-	-	1	-	1	-		-	-	-	-	-
[37]	IG1	-	-	-	1	12	-	-	-	-	-	-	-	1	-	-	-	-	1	1	-	-	-	1	-	-	1	-	-
	IG2	-	-	-	1	12	-	-	-	-	-	-	-	1	-	-	-	-	1	1	-	-	-	-	-	-	-	-	-
[38]	IG1	-	-	1	-	12	-	-	-	-	-	-	-	-	1	-	-	-	1	-	-	-	-	-	1	-	1	-	-
	IG2	-	-	1	-	12	-	-	1	1	-	1	-	-	-	1	-	-	1	-	1	-	-	-	-	-	-	-	-
[39]	IG1	-	-	1	-	6	1	-	1	1	1	-	-	-	-	1	-	1	-	1	1	-	-	-	1	-	1	1	-
	IG2	-	-	1	-	6	1	-	1	1	1	1	-	-	-	1	-	1	-	1	1	-	-	-	-	-	-	-	-
[40]	IG	-	-	1	-	12	1	-	1	1	1	-	1	-	-	1	-	1	1	1	1	-	1	-	1	-	1	-	-
[41]	IG1	-	-	1	-	12	1	-	-	-	1	1	-	-	-	-	-	1	1	1	-	1	-	-	-	1	1	1	-
	IG2	-	-	1	-	12	1	-	-	-	1	1	-	-	-	-	-	1	1	1	-	1	-	-	-	-	-	-	-
[42]	IG	-	-	1	-	6	1	-	1	-	1	-	-	1	-	-	1	-	-	1	-	-	-	1	1	-	-	-	1
[43]	IG	-	-	1	-	6	1	-	1	-	-	1	-	1	-	1	-	-	-	1	1	-	-	-	-	1	-	-	1
[44]	IG1	-	-	-	1	24	-	1	1	1	1	-	-	1	-	-	-	1	1	-	1	-	-	-	-	1	-	-	1
	IG2	-	-	-	-	24	-	1	1	-	1	-	-	-	-	-	-	-	1	-	1	-	-	-	-	-	-	-	-
[45]	IG	-	-	1	-	2.5	-	-	-	1	-	-	-	1	-	-	1	-	1	-	1	-	-	-	-	-	1	-	-
[46]	IG	-	-	1	-	6	-	1	1	-	-	-	-	-	-	-	-	1	1	1	1	-	1	-	-	-	1	-	-
[47]	IG	-	-	1	-	6	1	-	1	1	1	1	-	-	-	1	-	-	1	1	1	-	-	-	-	1	1	1	-
[48]	IG	1	-	-	-	4	1	-	-	-	-	-	-	-	1	1	-	-	-	1	1	-	-	-	1	-	-	-	1
[49]	IG1	1	-	-	-	5	1	-	-	-	-	1	-	-	-	1	-	-	-	1	-	1	-	-	-	-	1	-	-
	IG2	-	-	1	-	5	1	-	-	-	-	1	-	-	-	-	-	1	-	1	-	1	-	-	-	-	-	-	-
[50]	IG	-	-	-	1	3	1	-	-	-	-	-	-	1	-	1	-	-	-	1	1	-	1	-	1	-	-	-	1
[51]	IG1	-	1	-	-	2	1	-	-	-	-	-	-	-	-	-	1	-	-	-	-	-	-	-	-	1	1	1	-
	IG2	-	1	-	-	2	1	-	1	-	-	-	-	-	-	-	-	-	1	-	-	-	-	-	-	-	-	-	-
	IG3	-	1	-	-	2	1	-	1	-	-	-	-	-	-	-	1	-	1	-	-	-	-	-	-	-	-	-	-
[52]	IG	-	-	1	-	3	-	-	-	-	-	-	-	1	-	1	-	-	-	1	1	-	-	-	1	-	-	-	1
[53]	IG	-	-	-	1	12	-	-	1	1	1	1	-	-	-	1	-	1	-	1	1	-	-	-	-	1	1	-	-
[54]	IG	-	-	-	1	12	1	-	-	-	-	-	-	1	-	-	-	-	-	1	1	-	-	-	-	1	-	-	1
[55]	IG1	-	-	-	1	6	1	-	-	-	-	-	-	1	-	1	-	-	-	1	-	1	-	-	1	-	1	-	-
	IG2	-	-	-	1	6	1	-	-	-	-	-	-	-	-	1	-	-	-	-	-	1	-	-	-	-	-	-	-

Ref. reference number, 1: specified data, (-): No specified data, **Group:** IG, Intervention group. **Participants:** P, Parents; C, Children; P-C, Parents and children; FAM, Family. **Intervention characteristics:** Dur, Duration; Tra, Tracing. **Professional responsible:** Nur, Nurse; Nut, Nutritionist; PA, Physical activity professional; MD, Medical doctor; Psy, Psychologist; SW, Social worker; NS, No specified. **Intervention:** Phys. Act, Physical activity; Rec, Recommendations; PEP, Physical exercise program; **RA, Recreational activities**; Beh. Ter., Behavioral therapy; Cond, Conduct; NP, Nutrition plan; **RA, Recreational activities. Primary outcome:** BMI (kg/t2), Body mass index; BMI (z Body mass index z-score; BMI (SD), Body mass index standard deviation. **Main outcome:** SC, Significant changes; MT, Maintenance, NSC, No significant changes.

**Table 4 ijerph-18-03098-t004:** Statistically significant effects of the interventions lasting 6, and 12 months.

	Duration of the Intervention: 6 and 12 Months							
	Physical Activity			Education	Behavioral Therapy	Nutrition		
Statistically significant change over the primary outcomes	Recreational activities	Recommendations	Physical exercise program			Nutrition program	Recommendations	Cooking class
≤6 months	100%	59%	83%	100%	65%	100%	63%	50%
≥12 months	67%	100%	100%	92%	91%	100%	75%	100%

## Data Availability

The datasets used and/or analyzed during the current study are avail-able from the corresponding author on reasonable request.

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
