# Peer review of "Characteristics of Multicomponent Interventions to Treat Childhood Overweight and Obesity in Extremely Cold Climates: A Systematic Review of a Randomized Controlled Trial"

_ijerph, 2021, doi:10.3390/ijerph18063098_

Round 1

Reviewer 1 Report

The work would have a contribution to the field. Please consider the following suggestions to improve the manuscript in terms of clarity and rigor.

Abstract:

Should clarify the number of studies in addition to the number of articles

Mean adherence to program = 86%? What percentages for a range of different components, as different components may have different adherence?

72% of the interventions presented significant improvements in terms of what?

57% of the interventions were based on four components---how may interventions were based on three/two/single components or more than four components?

The secondary variables studied were nutritional status and…----what are the primary variables studied?

Introduction:

Language suggestions:

In the first sentence, should be “World”

Interventions with the family and at school are considered an important strategy against obesity—should be “interventions with the family and school are considered as…”

….an association between extreme cold weather---should be …an association between extreme cold weather and childhood overweight or obesity

Methods:

Why not include studies before 2010?

Did the review include cluster-randomized controlled trial?

“Period: between January 2010 and January 2020” in 2.2 repeated the words in 2.1

For data extraction, should also extract the information of sample size

Please explicitly clarify the definition of “selection bias”, “performance bias”, “detection bias”, “attrition bias”, and “reporting bias”, which should be the definition showed in figure 2

For data synthesis strategy, why not use meta-analysis?

Figure 1

How may duplicates were screened?

Should be “period”, rather than “periode”

Should distinguish the number of “studies” from the number of “articles” in the last box

Figure 2

What the definition of “blinding of participants and staff”? Interventions of this type usually could not blind participants and staff.

Figure 3

The title of figure 3 is ambiguous, should be clarified, such as “the percentage of variables of ….used in the included studies”

Table 1

“study variables for intervention effects” were not consistent with Figure 3

Why shows “4)” and “5)” in the last two rows but not other rows in the table?

Author Response

Dear reviewer,

We appreciate the opportunity to revise and re-submit our manuscript. We trust that you will find the current version acceptable for publication.

Our manuscript has been changed according to the required recommendations. Modifications to the manuscript text are denoted by red highlight. Please find our responses to each reviewer's comments below.

Sincerely, the Authors.

Abstract:

COMMENT: Should clarify the number of studies in addition to the number of articles.

ANSWER: Thank you for the comment. We have reviewed the articles again and we confirm that the total number of articles is 29 and that each of them contains only one study.

COMMENT: Mean adherence to program = 86%? What percentages for a range of different components, as different components may have different adherence?

ANSWER: We appreciate the comment, it is of great relevance.

The adherence to the programs, depending on the type of the intervention, was:

1) Physical activity-nutrition: the mean adherence was 100%.

2) Behavioral-nutrition: the mean adherence was 86%.

3) Behavioral-Education: the mean adherence was 93%.

4) Physical activity-Behavioral: the mean adherence was 86%.

5) Physical activity-Behavioral –Nutrition: the mean adherence was 81%.

6) Physical activity-Education –Nutrition: the mean adherence was 87%.

7) Physical activity-Behavioral –Education-Nutrition: the mean adherence was 84%.

8) Education: the mean adherence was 93%.

Page 1 Abstract

Page 4, 3.2 Characteristics of the studies analyzed

COMMENT: 72% of the interventions presented significant improvements in terms of what?

ANSWER: Thanks for your suggestion. We changed the term "significant improvements" for "positive effects" since our manuscript is not a meta-analysis. 72% of the interventions reported positive effects in the reduction of the BMI. Page 1, Abstract

COMMENT: 57% of the interventions were based on four components---how may interventions were based on three/two/single components or more than four components?

ANSWER: Thank you for your clarification. We change the phrase to make it more understandable: "57% of the studies reported post-intervention results”, on page 1.

COMMENT: The secondary variables studied were nutritional status and…----what are the primary variables studied?

 ANSWER: Thank you for this comment. The primary variables studied were BMI, BMI Z-score BMI-SDS. We included it on page 1, in the Abstract.

Introduction:

Language suggestions:

COMMENT: In the first sentence, should be “World

ANSWER: Thank you for your clarification. We corrected it on the manuscript, on page 2.

COMMENT: Interventions with the family and at school are considered an important strategy against obesity—should be “interventions with the family and school are considered as…”

ANSWER: Thank you again for your clarification. We corrected it on the manuscript, on page 2.

COMMENT: ….an association between extreme cold weather---should be …an association between extreme cold weather and childhood overweight or obesity

ANSWER: According to your suggestion, we corrected in on the manuscript, on page 2.

 Methods:

COMMENT: Why not include studies before 2010?

ANSWER: We decided to identify studies since 2010 taking as reference the World Health Organization (WHO), which established the objective of not increasing the obesity levels since 2010 for preschool children, adolescents and, adults. The WHO reported the Global Action Plan for the Prevention and Control of NCDs: 2013-2020 in Geneva, 2013.

COMMENT: Did the review include cluster-randomized controlled trial?

ANSWER: Thank you for your question. In this review, we only included randomized controlled trials.

COMMENT: “Period: between January 2010 and January 2020” in 2.2 repeated the words in 2.1

ANSWER: Thank you for your clarification. We corrected the phrase, on page 3, section 2.2.

COMMENT:  For data extraction, should also extract the information of sample size

ANSWER: According to your suggestion, we extracted the information of simple size, which was n= 4.434 subjects. You will find it on data extraction and on Table II, on page 4, section 2.3.  

COMMENT:  Please explicitly clarify the definition of “selection bias”, “performance bias”, “detection bias”, “attrition bias”, and “reporting bias”, which should be the definition showed in figure 2

ANSWER: According to your suggestion, we clarified the definition of such bias, extracted from the Cochrane Tool “Cochrane Handbook for Systematic Reviews of Interventions”, on page 18, figure 2.

Higgins, JPT; Green, S. (Eds.).  Cochrane Handbook for Systematic Reviews of Interventions. 5.1.0. Version. The Cochrane Collaboration, 2011. Available online: www.cochrane-handbook.org

  • "Selection bias": selection of participants in the sample based on a random process.
  1. Sequence generation: Random assignment allows the sequence to be unpredictable. An unpredictable sequence, combined with concealment of the allocation sequence, is sufficient to avoid selection bias. The random sequence can be done through computer software.
  2. Allocation concealment: Participants and recruiters do not know which group they belong to before the study begins. Allocation concealment attempts to avoid selection bias in the allocation to the intervention by protecting the sequence of allocation before and until allocation, and can always be successfully implemented regardless of the subject of study. (Schulz 1995b, Jüni 2001).
  • “Performance bias”: “Performance bias” refers to the process by which participants, operators and, researchers, including people evaluating the results, are not aware of the assignments of the interventions after the participants were included in the study.
  • “Detection bias”: It refers to the blinding of the evaluators of the results. It refers to the systematic differences between groups in the way in which the results were obtained.
  • “Attrition bias”: It refers to the differences between groups due to dropouts from the study. They can be voluntary or involuntary. An example is the exclusions where some participants are omitted from the analyzes, even though the data on their results is available.
  • "Reporting bias": Refers to the differences between the results presented and those not presented. In many cases, researchers tend to vary the focus of the studies because the results were not as expected.

 COMMENT:  For data synthesis strategy, why not use meta-analysis?

ANSWER: Thank you for your question. Our objective was to determine the characteristics of the interventions, but not the size effect. Page 18, figure 1.

http://scielo.isciii.es/scielo.php?script=sci_arttext&pid=S1134-80462014000600010

 COMMENT:  How may duplicates were screened?

ANSWER: In the data extraction, we used the Mendeley tool, with the fonction “remove duplicates”, and no duplicates were found.  Page 18, Figure 1.

COMMENT:  Should be “period”, rather than “periode”

ANSWER: Thank you for your clarification. We corrected on the manuscript on figure 1.

COMMENT:  Should distinguish the number of “studies” from the number of “articles” in the last box

ANSWER: Thank you again for your clarification. We changed on the manuscript on page 18, figure 1.

COMMENT:  What the definition of “blinding of participants and staff”? Interventions of this type usually could not blind participants and staff.

 ANSWER: Thank you for your clarification. “Blinding of participants and staff” refers to the process by which participants and researchers, including people evaluating results, are unaware of the allocations of interventions. As you suggest, in this type of intervention it is very difficult to blind participants and staff.

COMMENT:  The title of figure 3 is ambiguous, should be clarified, such as “the percentage of variables of ….used in the included studies”

 ANSWER: According to your suggestion, we changed the title of figure 3 to make it clearer, on page 19.

COMMENT:  “study variables for intervention effects” were not consistent with Figure 3

ANSWER: Thank you for your suggestion. We revised the variables of figure 3 and added a description. “Percentage of variables of a) Nutritional status (BMI, Body Mass Index), b) Physical condition and health (PAL, Physical Activity Level), and c) Psychological health, (HrQoL, Health-related quality of life), in the studies included, on page 19, Figure 3.

COMMENT:  Why shows “4)” and “5)” in the last two rows but not other rows in the table?

ANSWER: We corrected it on table I, adding the other rows, on page 21, Table I.

Reviewer 2 Report

It is a clear and coherent work. It has limitations that do not prevent its important contribution to the knowledge of such an important subject for children's health.

Regarding the strengths of article ‘Characteristics of multicomponent interventions to treat childhood overweight and obesity in extremely cold climates: A systematic review of randomized controlled trial’, I can highlight the importance of the subject (children's health) as it is a current and growing problem worldwide. I also think that the title correctly describes its content, its appropriate abstract, its key words and its bibliography, that are sufficiently up to date for a research like this, in my opinion. 

On their weaknesses I can point out that the research analyses very different studies that followed different methodologies, so the conclusions drawn are valid only as a general guide to the subject, as the paper rightly acknowledges. I also consider a weakness that the BMI was used as a reference even though it is not the one most recommended by the WHO. As well as the risk of detection bias, there was a 66% "unclear risk" in the results.

Small mistakes to correct:

Page 2, line 2: world Health Organization. W not in capital letter.

Page 2 (last paragraph) and page 3 (2.1. - Search strategy to identify studies) Norway appears twice in the country lists.

In any case, I believe that its contributions to such an important topic based on so many studies make it worthy of publication. 

Author Response

Dear reviewer,

We appreciate the opportunity to revise and re-submit our manuscript. We trust that you will find the current version acceptable for publication.

Our manuscript has been changed according to the required recommendations. Modifications to the manuscript text are denoted by red highlight. Please find our responses to each reviewer's comments below.

Sincerely, the Authors.

COMMENT:  Page 2, line 2: world Health Organization. W not in capital letter.

ANSWER: Thank you for your clarification. We corrected it on the manuscript, on page 2.

COMMENT:  Page 2 (last paragraph) and page 3 (2.1. - Search strategy to identify studies) Norway appears twice in the country lists.

ANSWER: We thank you again for you clarification. We corrected it on pages 2 and 3.

In any case, I believe that its contributions to such an important topic based on so many studies make it worthy of publication. 

Round 2

Reviewer 1 Report

The manuscript has been improved and it's better to further consider the following suggestion.

It seems that the conclusion "the interventions that were effective in reducing childhood overweight and obesity were those that combined the components of physical activity, diet, education.." cannot be logically concluded from the description of the results in the abstract as well as in the main body of the manuscript (especially for interventions which duration equal or more than twelve months).

Author Response

Comments:

It seems that the conclusion "the interventions that were effective in reducing childhood overweight and obesity were those that combined the components of physical activity, diet, education.." cannot be logically concluded from the description of the results in the abstract as well as in the main body of the manuscript (especially for interventions which duration equal or more than twelve months).

Dear Reviewer,

We appreciate your suggestions. Our manuscript has been changed according to the required recommendations, and changes are highlighted in yellow.

In response to the reviewer, we point out that the main results and effects of the interventions are now detailed in the abstract (results and conclusion) and the full text (results, discussion, and conclusion).

We hope that these improvements contribute to the progress of the review of the manuscript.

Kind regards,

Rafael Zapata Lamana.
